# The Hormetic Effect Observed for Benzalkonium Chloride and Didecyldimethylammonium Chloride in *Serratia* sp. HRI

**DOI:** 10.3390/microorganisms11030564

**Published:** 2023-02-23

**Authors:** Samantha J. McCarlie, Laurinda Steyn, Louis L. du Preez, Charlotte E. Boucher, Julio Castillo Hernandez, Robert R. Bragg

**Affiliations:** 1Department of Microbiology and Biochemistry, University of the Free State, Bloemfontein 9301, South Africa; 2eResearch & HPC: ICT Services, University of the Free State, Bloemfontein 9301, South Africa

**Keywords:** quaternary ammonium compounds, antimicrobial tolerance, growth analysis

## Abstract

Hormesis, or the hormetic effect, is a dose- or concentration-dependent response characterised by growth stimulation at low concentrations and inhibition at high concentrations. The impact of sub-lethal levels of disinfectants on the growth of *Serratia* species is critical to understanding the increasing number of outbreaks caused by this pathogen in healthcare settings. *Serratia* sp. HRI and *Serratia marcescens* ATCC 13880 were cultivated in sub-lethal levels of benzalkonium chloride (BAC), Didecyldimethylammonium chloride (DDAC), and Virukill^TM^. The maximum specific growth rates, doubling times, and cell counts were compared. The results revealed significant increases in maximum specific growth rates and shorter doubling times for *Serratia* sp. HRI when cultivated in sub-lethal levels of BAC and DDAC. The significant stimulatory effect of sub-lethal levels of these disinfectants for *Serratia* sp. HRI represents the first time hormesis has been observed in a Gram-negative bacterium for any disinfectant. Furthermore, this study is the first to observe the hormetic effect after treatment with DDAC and the second study to date analysing the impact of sub-lethal levels of disinfectants on the growth of bacterial species.

## 1. Introduction

Growth analysis of *Serratia* species has focused on the growth kinetics and associated production of the red pigment, prodigiosin, by members of the *Serratia marcescens* species [1]. Prodigiosin production is associated with a faster growth rate and increased biomass due to coupled ATP production in pigment-producing strains of *Serratia* species (such as *Serratia marcescens* ATCC 13880) [2]. In contrast, non-pigmented strains (such as *Serratia* sp. HRI) are associated with a lower growth rate and cell density over time compared to pigmented strains. Little information exists on the effect of disinfectants on the growth of *Serratia* species. As a nosocomial pathogen increasingly associated with outbreaks in healthcare facilities [3,4,5,6,7], the impact of sub-lethal levels of disinfectants on the growth of this pathogen is crucial to direct hygiene and safety standards in the future.

As early as 1981, *S. marcescens* was identified as the cause of nosocomial outbreaks, often isolated from the same bottles of disinfectants or antiseptics responsible for preventing these outbreaks [8]. *S. marcescens* was isolated from a 2% chlorhexidine solution after 27 months of storage [8]. Not only was this isolate able to survive for prolonged periods in the disinfectant solution, but it was able to survive in concentrations up to 20 g/mL, 20 times higher than the minimal inhibitory concentration (MIC) value (1024 g/mL) recorded for this isolate [8].

In terms of Quaternary Ammonium Compound (QAC) exposure at sub-MIC levels, current literature shows either no significant effect or an inhibitory effect on bacterial growth rate and cell density [9,10]. Recent studies found no significant difference in growth rates for *Escherichia coli* and *Listeria monocytogenes* respectively, after exposure to sub-MIC levels of benzalkonium chloride (BAC) when compared to unchallenged controls [9,11]. Whereas for *S. marcescens*, cell density decreased for both resistant and susceptible strains, even at the lowest concentration of BAC [12]. This phenomenon has also recently been observed in *Bacillus cereus* and *Escherichia coli*, where the addition of sub-MIC levels of Didecyldimethylammonium chloride (DDAC) resulted in a reduction in optical density (OD) and CFU/mL values coupled with a lower growth rate compared to positive controls (no disinfectant added) [10]. In contrast, significant stimulation of growth after the addition of disinfectants at a low concentration has been observed in one instance by *Staphylococcus aureus* [13]. The hormetic effect is the growth stimulation by adaptive responses triggered by low concentrations of antimicrobials [13,14]. This phenomenon has been associated with various antibiotics and metal nanoparticles [14,15,16,17,18], but limited information exists on whether this applies to disinfectants and, if so, which active ingredients. hormesis was observed for *S. aureus*, where a significant increase in growth rate was recorded for low chlorhexidine digluconate, povidone iodine, and BAC concentrations. A similar trend of increased growth rate was observed for *Pseudomonas aeruginosa* for all three disinfectants at low concentrations, but the changes were insignificant. This work suggests that the hormetic effect does apply to disinfectants, but it has only been confirmed in *S. aureus* and no other bacterial species yet.

The impact of the hormetic effect is far-reaching in any industry that relies upon disinfection and cleaning protocols. In areas with limited accessibility (air vents), high organic load (surgical wounds), or biofilms (infant feeding bottles) [3], hormesis could have devastating effects. The cleaning and disinfection of these areas and products with antiseptics or disinfectants at incorrect concentrations could cause stimulation of microbial growth instead of an inhibitory effect [2]. Therefore, the hormetic effect could have severe and far-reaching consequences for healthcare systems and infection control.

This study compared the maximum specific growth rate and doubling times of *Serratia marcescens* ATCC 13880 type strain (pigment-producing) and the antimicrobial-resistant isolate *Serratia* sp. HRI (non-pigment producing) at sub-lethal levels of three QAC-based compounds benzalkonium chloride (BAC), didecyldimethylammonium chloride (DDAC), and Virukill^TM^. To date, some work has been done on the effect of either DDAC or BAC on the growth of bacteria [9,10,11,13]; however, to our knowledge, this study is the first to compare the effects of different QAC-based disinfectants to each other on the same microorganism. In addition, little to no research has been done on the effect of multiple QAC-based disinfectants on *Serratia* species’ growth. An important question is whether disinfectants induce the hormetic effect. This study is the first to confirm a significant hormetic effect induced by these disinfectants for a bacterium other than *Staphylococcus aureus*, and thus is the second study to confirm hormesis after exposure to disinfectants. Furthermore, this is the first time a significant hormetic effect has been observed in a Gram-negative bacterium after treatment with disinfectants.

## 2. Materials and Methods

### 2.1. Microorganisms 

The microorganisms used in this study were *Serratia* sp. HRI, isolated from a bottle of DDAC-based disinfectant [19], and the closest related type strain, *Serratia marcescens* subsp. marcescens ATCC 13880, obtained from the American Type Culture Collection.

### 2.2. Disinfectants

Benzalkonium chloride obtained from Sigma-Aldrich (≥50% in water), Didecyldimethylammonium chloride (DDAC) obtained from ICA International Chemicals (80% UNIQUAT), and Virukill™ (Poly Dimethyl Ammonium Chloride 120 g/L). The MIC levels of these disinfectants for *Serratia* sp. HRI and *Serratia marcescens* subsp. marcescens ATCC 13880 were previously reported [19], and sub-MIC levels for this study were between 10% and 50% of the MIC level for each bacteria.

### 2.3. Inoculum and Shake Flask Preparation

Inocula of *Serratia* sp. HRI and *S. marcescens* ATCC 13880 were prepared by transferring a single colony from 24 h agar plates to 25 mL LB medium (10 g/L Merck Tryptone, 10 g/L Merck sodium chloride, 5 g/L Merck yeast extract) in 250 mL Erlenmeyer flasks and incubated at 37 °C for 16 h. From this pre-inoculum, 2.5 mL was added into 500 mL Erlenmeyer flasks containing LB media supplemented with one of the three QAC-based disinfectants (benzalkonium chloride, Virukill™, and DDAC) and incubated at 250 r min^−1^ and 37 °C. The growth analysis of *Serratia marcescens* ATCC 13880 was performed at a sub-MIC level of 0.0001% (final volume) for all three disinfectants for three biological replicates. The sub-MIC levels of disinfectant for *Serratia* sp. HRI were 0.01% (final volume) for each disinfectant during growth analysis. After adding the disinfectant to the LB media (before inoculation), a blank reading was taken and subtracted from consequent optical density (OD) measurements. Blank readings for each sample were taken after the disinfectant was added, as the high amount of disinfectant in the media impacted the OD reading. Subsequently, an absorbance spectrum was generated for the three QAC-based disinfectants, as depicted in Figure A1.

Three controls were added, the first comprising each bacterium in growth media without disinfectant; the next control was included with no inoculum or disinfectant to rule out any contamination; and the last control contained disinfectant but no inoculum to ensure any change in absorbance over time was due to the culture and not interactions of the disinfectant. Three biological replicates were done for each experimental condition.

### 2.4. Analytical Methods

Optical density was measured with a PhotoLab S6 spectrophotometer at 690 nm. The culture was diluted when the OD exceeded 0.8. The maximum specific growth rate and doubling time of a batch culture were determined by linear regression analysis with Microsoft Excel (Microsoft Corporation, WA, USA) of the exponential phase of the growth curve. Multiple paired t-tests were performed (Paired Two Sample for Means) to determine whether there were significant differences in each bacterium’s maximum specific growth rates. A *p*-value (P(T ≤ t) two-tail) less than 0.05 indicated a statistically significant difference with 95% confidence.

### 2.5. 60-Min Growth Analysis with Cell Count

Samples were prepared and run as indicated above with sub-MIC levels of disinfectant. At 10-min intervals from time zero, OD (690 nm) readings were taken for each sample until 60 min. In addition to an OD measurement, cell counts were done at each time point. Samples were serially diluted in sterile water, and dilutions at the 10^−6^ and 10^−7^ levels were plated out on TSA agar and incubated overnight at 37 °C. After that, CFU/mL was calculated for each sample at each time point. An ANOVA single factor test (Microsoft Corporation, Redmond, WA, USA) was run to determine whether there was a significant difference between the CFU/mL values at each time point for each biological replicate.

### 2.6. Absorbance Spectrum

QAC disinfectants were prepared at 0.01% in LB media, the highest concentration used for growth analysis. Samples were loaded onto a 96-well plate and analysed using a SpectraMax M2 microtitre plate spectrophotometer, where a range of wavelengths from 200 to 1000 nm were scanned for peaks in absorbance due to the presence of the disinfectant. This was done to determine whether changes in OD at 690 nm could be due to interfering absorbance of the disinfectant molecules and not attributable to the inoculum. The data was uploaded to SoftMaxPro (v. 5.2) and shuffled (blank deducted) before being plotted as an absorbance spectrum in Figure A1.

## 3. Results

### 3.1. Growth Analysis of Serratia sp. in the Presence of Sub-MIC Level Disinfectants

When cultivated in non-supplemented media, *Serratia marcescens* ATCC 13880 type strain had a lower doubling time and a significantly higher maximum specific growth rate compared to the resistant isolate, *Serratia* sp. HRI is depicted in Table 1.

After adding benzalkonium chloride to the medium, the maximum specific growth rate of the type strain decreased slightly, and the correlating doubling time increased, as expected after adding an antimicrobial. However, adding BAC to the medium resulted in a significantly higher maximum specific growth rate for *Serratia* sp. HRI compared to non-supplemented media depicted in Figure 1.

Similar results were observed after adding DDAC to the medium; for the type strain, the growth rate decreased slightly (Figure 2), but no significant changes were observed. In contrast, sub-MIC levels of DDAC resulted in a significant increase in the maximum specific growth rate for *Serratia* sp. HRI compared to non-supplemented media (Table 1). Furthermore, as with BAC, the growth rate of *Serratia* sp. HRI increased to surpass the type strain after the addition of DDAC. This correlated to a shorter doubling time for the resistant isolate and matched the results after adding BAC to the medium (Table 1).

After adding the more recently developed QAC formulation, Virukill™, the results did not follow the trends observed with the older generation disinfectants (BAC and DDAC). After adding Virukill™, as with the control (no disinfectants added to the growth media), the type strain had a significantly higher maximum specific growth rate and a lower doubling time when compared to the resistant isolate depicted in Figure 3. There was also no significant difference between growth in sub-MIC levels of Virukill and the control group of *Serratia* sp. HRI cultured without a disinfectant (Table 1).

Interestingly, in the growth curves of *Serratia* sp. HRI after the addition of BAC (Figure 1) and DDAC (Figure 2) a drop in absorbance was observed within the first 30 min of cultivation. This drop in OD was not present for the type strain and did not occur in the positive control of *Serratia* sp. HRI in each experiment. The decrease in absorbance was only observed for the resistant strain when BAC or DDAC was added to the media.

### 3.2. Growth and Cell Count Analysis of Serratia sp. HRI and Serratia Marcescens ATCC 13880

It was speculated that the drop in OD at the 30-min mark could be due to cell death after the addition of disinfectants. To test this, growth analyses were repeated, represented by Figure 4, focusing on the first hour of growth and calculating CFU/mL values to determine if the drop in OD could be correlated to a decline in cell count.

Figure 4 indicates that the drop in OD for BC and DDAC seemed to be at its lowest at 20 min before a gradual rise in absorbance started. The CFU/mL values calculated showed no correlation to the drop in absorbance within the first hour of cultivation, which suggests that a drop in cell count is not responsible for the decline in OD readings observed for *Serratia* sp. HRI.

### 3.3. Absorbance Spectrum of QAC-Based Disinfectants

To account for any changes in absorbance caused by the disinfectant molecules, an absorbance spectrum was generated for the three QAC-based disinfectants. As depicted in Figure A1, absorbance was high, being between 200 nm and 300 nm for all three disinfectants. For growth analysis, readings were taken at 690 nm, where absorbance readings for the disinfectants at 0.01% were as follows: Virukill™ at 0.0464, DDAC at 0.1013, and BC at 0.1433.

## 4. Discussion

Previous studies on this pathogen have shown that unchallenged pigment-producing strains of *Serratia* species have higher ATP production, cell yield, and specific growth rate compared to strains that do not produce the prodigiosin pigment [2]. This was observed in this study when comparing the maximum specific growth rate and doubling time of *Serratia marcescens* ATCC 13880 (pigment-producing) and *Serratia* sp. HRI (non-pigmented). The lower growth rate of the resistant isolate observed in Table 1 (*Serratia* sp. HRI) is also expected as a lower growth rate can contribute to decreased susceptibility against antimicrobials that target cell membrane synthesis, such as β-lactam antibiotics [20,21,22]. This also applies to disinfectants, as a lower growth rate aided in reduced susceptibility to benzalkonium chloride in *Pseudomonas aeruginosa* [23]. This indicates that *Serratia* sp. HRI may have evolved a slower growth rate compared to *Serratia marcescens* ATCC to increase tolerance to some antimicrobials [24].

When analysing growth patterns of the type strain, *Serratia marcescens* ATCC 13880, adding sub-MIC levels of BAC, DDAC, and Virukill™ disinfectants had no significant impact on the growth rate, as illustrated in Figure 1, Figure 2 and Figure 3. In contrast, after sub-MIC levels of BAC and DDAC were added to the resistant isolate, *Serratia* sp. HRI, the specific growth rate increased significantly in response to both disinfectants compared to unsupplemented media (Table 1). An increase in maximum specific growth rate and cell density after the addition of disinfectants has been recorded once before in *Staphylococcus aureus* exposed to chlorhexidine digluconate, BAC, and povidone iodine [13]. This was the first time the hormetic effect was observed for disinfectant solutions.

The results for Virukill™, illustrated in Table 1, were different from earlier generations of QACs, BAC, and DDAC when added during the cultivation of *Serratia* sp. HRI. Although the active ingredient of Virukill™ is similar to DDAC, unknown additives in this product could be responsible for the extraordinary result. No significant differences were observed after adding Virukill^TM^ during bacterial cultivation, which suggests that new formulations of QAC-based disinfectants do not induce the hormetic effect. This is vitally important in designing and producing new antimicrobial compounds and warrants further study.

Upon closer inspection of the BAC and DDAC growth curves, it was noted that during the cultivation of *Serratia* sp. HRI, there was a drop in absorbance at around the 20-min mark, which evened out again after 50 min (Figure 4). This drop was not present in the growth analysis of *Serratia marcescens* ATCC 13880 and was absent from the *Serratia* sp. HRI positive control; it is only seen when *Serratia* sp. HRI is cultivated in media with sub-MIC levels of BAC (Figure 1) and DDAC (Figure 2). The cell counts depicted in Figure 4 show that the drop in absorbance does not correlate with a drop in CFU/mL values. The negative control containing disinfectant also did not follow this pattern indicating that it is not the disinfectant sticking to the glassware or precipitating out of the media. After adding the disinfectant to the media (0.1%), the absorbance readings increased by about 0.2 units; therefore, if the disinfectant was removed from the media by cell interactions, it would have a significant enough impact to affect the absorbance readings.

The initial drop in absorbance coupled with an increased maximum specific growth rate for *Serratia* sp. HRI has not been observed and reported in the current literature. This brings forward a hypothesis that explains both the drop in absorbance readings within the first 30 min of growth and the increase in the maximum specific growth rate of *Serratia* sp. HRI. The action of QAC-based disinfectants could explain the initial drop in absorbance readings. QAC-based disinfectants bind to the negatively charged cell membranes of bacteria and damage the cell wall [25,26]. This can result in an initial decrease in absorbance readings as the disinfectant is no longer suspended in the media. Thereafter, the increased growth rate could be due to *Serratia* sp. HRI using the disinfectant as an energy source. Bacteria can metabolise QAC-based disinfectants, as observed in *Pseudomonas* species [27,28]. Additionally, several articles have reported the isolation of *Serratia marcescens* from bottles of antiseptics and disinfectants after years of storage without any additional carbon or nitrogen source [8,12,29,30]. The resistant isolate used in this study, *Serratia* sp. HRI, was isolated from a bottle of DDAC-based disinfectant diluted in water, where no other energy sources were present [19]. This study suggests that this isolate can metabolise benzalkonium chloride and DDAC. These antimicrobial compounds may be metabolised into less toxic intermediates and biomass, accounting for the increased growth rate of *Serratia* sp. HRI. The use of QAC-based disinfectant compounds as an energy source elucidates alternative bacterial metabolism pathways, allowing pathogenic bacteria to use antimicrobial compounds as their sole carbon and energy source [31]. Metabolic adaptions include hydroxylation, N-dealkylation, N-demethylation, and β-oxidation of QAC compounds [31,32,33,34]. The metabolism of disinfectants has not yet been observed for *Serratia* species, and future work will be needed to confirm if these compounds are being metabolised or degraded.

The hormetic effect is hypothesised to be due to an overreaction of the bacterial stress response creating a stimulatory effect [35]. This hypothesis is well-founded as these microorganisms are certainly under stress after treatment with antimicrobials. It is interesting to note that this work may provide a more specific mechanism of action of the hormetic effect, which is the metabolism of antimicrobial compounds as an energy source resulting in increased biomass. Whether the metabolism of these compounds is linked to the stress response remains unknown. Nevertheless, this work presents a possible explanation of how the hormetic effect stimulates growth and reiterates the importance of the stress response in bacterial tolerance and resistance. Further work will be required to elucidate whether the metabolism of these compounds plays a role in hormesis and if this is linked to the stress response.

## 5. Conclusions

The results of this study represent only the second time a significant hormetic effect has been confirmed for disinfectants and the first time a hormetic effect has been observed for disinfectants in a Gram-negative bacterium. This is the first time DDAC has induced the hormetic effect. Interestingly, a hormetic effect was not induced by the more recently developed generation QAC (Virukill^TM^), suggesting that it is possible to design products that inhibit this effect. This implication could direct the design of new antimicrobials and indicates that there may be a solution to this devastating effect. Furthermore, this study provided insight into cell-disinfectant interactions elucidated by changes in absorbance and may provide the first evidence of the metabolism of disinfectants by *Serratia* species.

*Serratia marcescens* is an opportunistic pathogen increasingly associated with infections and outbreaks in neonatal and paediatric healthcare units [3,4,5,6,7]. Healthcare facilities use disinfectant products daily for infection control, wound care, and to prevent the development of antimicrobial resistance. The observations in this study suggest that incorrectly applied or diluted disinfectant products can have the opposite effect as intended; instead of inhibiting microbial growth, it can be stimulated. In healthcare settings where antimicrobial resistance is a major issue, this could have devastating impacts and lead to increased pathogenic outbreaks. Special care must be taken in areas where microbial growth needs to be controlled, such as operating theatres, wound care, catheters, and surgical instruments. The observations made in this study will be vitally important in preventing future hospital acquired infections and infection control world-wide.

## Figures and Tables

**Figure 1 microorganisms-11-00564-f001:**
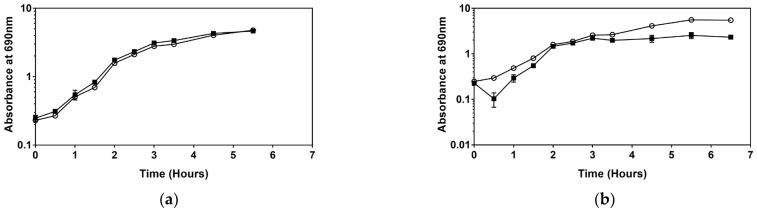
Growth comparison of supplementation with sub-MIC levels of benzalkonium chloride (■) and without BAC (○). For some points, the error bars are shorter than the height of the symbol. In these cases, the error bars were not drawn: (**a**) *Serratia marcescens* ATCC 13880; (**b**) *Serratia* sp. HRI.

**Figure 2 microorganisms-11-00564-f002:**
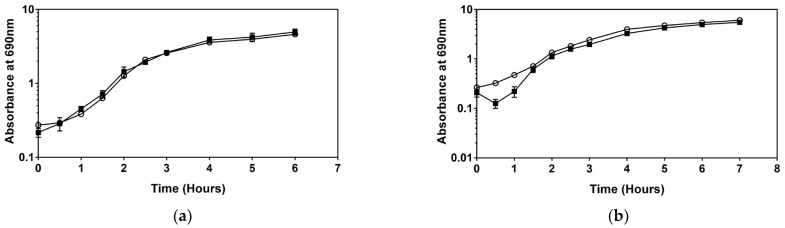
Growth comparison of supplementation with sub-MIC levels of DDAC (■) and without DDAC (○): (**a**) *Serratia marcescens* ATCC 13880; (**b**) *Serratia* sp. HRI.

**Figure 3 microorganisms-11-00564-f003:**
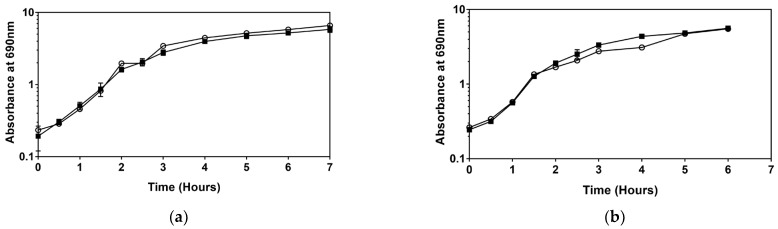
Growth comparison of supplementation with sub-MIC levels of Virukill (■) and without Virukill (○): (**a**) *Serratia marcescens* ATCC 13880; (**b**) *Serratia* sp. HRI.

**Figure 4 microorganisms-11-00564-f004:**
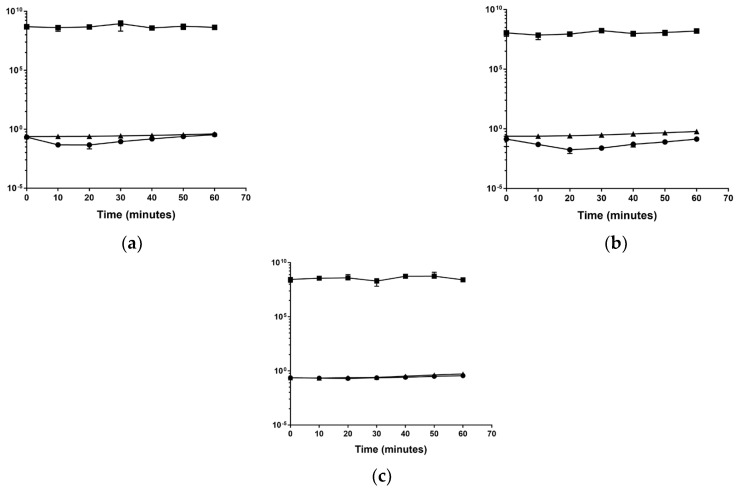
Comparison of changes in absorbance readings (●) and CFU/mL values for *Serratia* sp. HRI (■) as well as CFU/mL values of *Serratia marcescens* ATCC 13800 (▲) over 60 min cultivated in the presence of sub-MIC levels of disinfectants: (**a**) Benzalkonium chloride; (**b**) DDAC; (**c**) Virukill.

**Table 1 microorganisms-11-00564-t001:** Calculated maximum specific growth rate (µmax h^−1^) and growth kinetics analysis of *S. marcescens* ATCC 13880 and *Serratia* sp. HRI without and after adding three disinfectants (Benzalkonium chloride, DDAC, and Virukill^TM^). A1-3: *S. marcescens* ATCC 13880 biological replicates exposed at 0.0001% of disinfectant; H1-3: *Serratia* sp. HRI biological replicates exposed at 0.01% of disinfectant; A/H+: positive control with no disinfectant added.

Sample	µmax (h^−1^)	Ave. µmax	Doubling Time (Td, h)	Ave. Td
**No Disinfectant**				
A1	1.06	1.07 ± 0.04	0.65	0.65
A2	1.04		0.67	
A3	1.11		0.62	
H1	0.93	0.92 ± 0.01	0.74	0.75
H2	0.91		0.76	
H3	0.92		0.76	
**Sub-MIC Level Benzalkonium Chloride**				
A1	1.06	1.04 ± 0.04	0.52	0.67
A2	1.00		0.73	
A3	1.07		0.76	
H1	1.61	1.75 ± 0.23	0.51	0.48
H2	1.62		0.51	
H3	2.01		0.41	
A+	0.96	N/A	0.72	N/A
H+	0.89	N/A	0.78	N/A
**Sub-MIC Level DDAC**				
A1	0.90	0.97 ± 0.01	0.77	0.7634
A2	1.08		0.73	
A3	0.93		0.79	
H1	1.44	1.43 ± 0.01	0.58	0.5172
H2	1.43		0.48	
H3	1.41		0.49	
A+	1.02	N/A	0.69	N/A
H+	0.84	N/A	0.82	N/A
**Sub-MIC Level Virukill™**				
A1	1.09	1.14 ± 0.05	0.64	0.6075
A2	1.16		0.60	
A3	1.18		0.59	
H1	0.93	0.94 ± 0.04	0.74	0.7353
H2	0.91		0.76	
H3	0.99		0.70	
A+	1.12	N/A		N/A
H+	0.94	N/A		N/A

## Data Availability

The data presented in this study are available on request from the corresponding author. The data are not publicly available due to privacy restrictions.

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
