# Peer review of "The Hormetic Effect Observed for Benzalkonium Chloride and Didecyldimethylammonium Chloride in Serratia sp. HRI"

_microorganisms, 2023, doi:10.3390/microorganisms11030564_

Round 1

Reviewer 1 Report

In the current study, authors have investigated the effect of three disinfectants Benzalkonium chloride, Didecyldime thylammonium chloride and Virukill at sub-lethal levels on the maximum specific growth rates and doubling time of Serratia sp. HRI and Serratia marcescens. Authors have confirmed a hormetic effect in the Serratia species. Most importantly, hormetic effect was not seen with Virukill which is a recently developed broad spectrum disinfectant endorsing its high potency. Overall study is well designed, the manuscript is well written with minor grammatical errors, and data justifies the conclusion.

Author Response

Reviewer comment: Overall study is well designed, the manuscript is well written with minor grammatical errors, and data justifies the conclusion.

Authors response: Language and grammatical modifications have been made throughout the manuscript.

Reviewer 2 Report

Dear Authors

I have read the submitted paper titled "Hormetic effect observed for benzalkonium chloride and Didecyldimethylammonium chloride in Serratia sp. HRI"

The topic is of interest but not novel, material and methods are sound, results well explained. I believe that this is a preliminary  study that should certainly be implemented with experiments on biofilm formation by stimulation at low concentration of disinfectants. In addition it would be interesting to evaluate the Hormetic effect also in other gram negative bacteria ( CR Acinetobacter baumanni, CR Klebsiella pneumoniae ) responsible for  many nosocomial outbreacks.

The mechanisms responsible for hormetic effect  should also be investigated.

I have only a suggestion

Introduction line 38 Serratia HRI   HRI is acronym  ? Please specify

Author Response

Reviewers comment: Introduction line 38 Serratia HRI, HRI is acronym ? Please specify

Authors response: Yes HRI was initially an acronym for "Highly Resistant isolate" however this isolate is in the process of being characterised and renamed. Unfortunately we are unaware of when the renaming process will be completed and as such we will continue to refer to it as Serratia sp. HRI. 

Reviewer 3 Report

Samantha J. McCarlie and co-authors present a quality and well-written experimental manuscript describing hormetic effect observed for benzalkonium chloride and didecyldimethylammonium chloride in Serratia sp. HRI.

Authors compared the maximum specific growth rate and doubling times of Serratia marcescens and the antimicrobial-resistant isolate Serratia sp. HRI (non-pigment producing) at sub-lethal levels of three QAC-based compounds benzalkonium chloride , didecyldimethylammonium chloride and Virukill. To date, some work has been done on the effect of either DDAC or BAC on the growth of bacteria, however, this study is the first to compare the effects of different QAC-based disinfectants to each other on the same microorganism. In addition, little to no research has been done on the effect of multiple QAC-based disinfectants on Serratia species’ growth. An important question is whether disinfectants induce the hormetic effect. 

Authors claim that this study is the first to confirm a significant hormetic effect induced by these disinfectants for a bacterium other than Staphylococcus aureus, and thus is the second study to confirm hormesis after exposure to disinfectants. Furthermore, this is the first time a significant hormetic effect has been observed in a Gram-negative bacterium after treatment with disinfectants.

Finally, authors conclude that this work puts forward a possible explanation of how the hormetic effect stimulates growth and reiterates the importance of the stress response in bacterial tolerance and resistance. Further work will be required to elucidate whether metabolism of these compounds play a role in hormesis and if this is linked to the stress response.

==============================

Other comments:

1) Please check for typos throughout the manuscript.

2) With regards to Serratia marcescens microorganism – authors are kindly encouraged to cite the following article that describes specific enzymatic activity of S. marcescens which is relevant for healthcare.
DOI: 10.3389/fphar.2018.00114

Author Response

Reviewers comment: Please check for typos throughout the manuscript.

Authors response: Additional language and grammatical checks have been done.

Reviewers comment:

With regards to Serratia marcescens microorganism – authors are kindly encouraged to cite the following article that describes specific enzymatic activity of S. marcescens which is relevant for healthcare.

DOI: 10.3389/fphar.2018.00114

Authors response: Noted, thank you